

# Fungal survival under temperature stress: a proteomic perspective

Nurlizah Abu Bakar[1,2], Saiful Anuar Karsani[3] and Siti Aisyah Alias[1,2]

[1] Institute of Ocean and Earth Sciences, Universiti Malaya, Kuala Lumpur, Malaysia
[2] National Antarctic Research Centre, Universiti Malaya, Kuala Lumpur, Malaysia
[3] Institute of Biological Sciences, Faculty of Science, Universiti Malaya, Kuala Lumpur, Malaysia

## ABSTRACT

**Background**. Increases in knowledge of climate change generally, and its impact on agricultural industries specifically, have led to a greater research effort aimed at improving understanding of the role of fungi in various fields. Fungi play a key role in soil ecosystems as the primary agent of decomposition, recycling of organic nutrients. Fungi also include important pathogens of plants, insects, bacteria, domestic animals and humans, thus highlighting their importance in many contexts. Temperature directly affects fungal growth and protein dynamics, which ultimately will cascade through to affect crop performance. To study changes in the global protein complement of fungi, proteomic approaches have been used to examine links between temperature stress and fungal proteomic profiles.

**Survey methodology and objectives**. A traditional rather than a systematic review approach was taken to focus on fungal responses to temperature stress elucidated using proteomic approaches. The effects of temperature stress on fungal metabolic pathways and, in particular, heat shock proteins (HSPs) are discussed. The objective of this review is to provide an overview of the effects of temperature stress on fungal proteomes.

**Concluding remarks**. Elucidating fungal proteomic response under temperature stress is useful in the context of increasing understanding of fungal sensitivity and resilience to the challenges posed by contemporary climate change processes. Although useful, a more thorough work is needed such as combining data from multiple -omics platforms in order to develop deeper understanding of the factor influencing and controlling cell physiology. This information can be beneficial to identify potential biomarkers for monitoring environmental changes in soil, including the agricultural ecosystems vital to human society and economy.

# INTRODUCTION

Global warming and associated environmental changes impact living organisms in various ways. The recent Intergovernmental Panel on Climate Change (*IPCC, 2018*) Special Report: Global Warming of 1.5 °C, predicted that extreme hot day events in mid-latitudes may warm by up to ∼3 °C with global warming of 1.5 °C. It is also predicted that extreme night-time low temperatures at high latitudes warm by ∼4.5 °C (*IPCC, 2018*). If current trends persist, global ice loss will continue, the temperate regions will experience more

Corresponding author
Siti Aisyah Alias, saa@um.edu.my

frequent extreme high and low temperature events (*Francis & Skific, 2015*), and different parts of the tropics will experience longer drought and wet seasons (*Overland et al., 2015*). This can potentially lead to the breakdown of ecosystem structure and functions, making them yet more sensitive to future climate changes. There have already been some drastic changes in rainfall patterns and patterns of temperature variation (*Kundzewicz et al., 2005*), alreadyleading to significant reductions in land surface area suitable for crop cultivation (*Lesk, Rowhani & Ramankutty, 2016*). Several reviews have addressed research approaches for the development of improved crops in low-quality soils, and of soil management systems to improve resilience to current climate change challenges (*Fan et al., 2012*; *Rao et al., 2016b*; *Zhang, Chen & Vitousek, 2013*). Major outcomes of these include the use of better-adapted phenotypes (*Rao et al., 2016a*; *Simova-Stoilova, Vassileva & Feller, 2016*), integrated soil-crop system management (*Zhang, Chen & Vitousek, 2013*), and maintenance of functional diversity in the soil fungal community by retaining a capacity for symbiosis-driven recycling of organic nutrient pools (*Kyaschenko et al., 2017*).

The fundamental importance of including mycological studies in efforts to sustain agricultural industries and underpin economic growth is recognised in the European Union's common research 2014–2020 program, known as *Horizon 2020* (*Lange et al., 2012*). Fungi are generally the most important agent of primary decomposition and nutrient cycling in soil microbial communities (*A'Bear, Jones & Boddy, 2014b*). In boreal forest management and long-term forest production, the functional diversity of the ectomycorrhizal fungal community plays an important role in facilitating symbiosis-driven recycling of organic nutrient pools (*Kyaschenko et al., 2017*). The key role of fungi identified in such studies highlights the importance of understanding fungal responses to temperature stress. Active fungi initiate stress tolerance strategies at the molecular level and this can be observed in the physiological changes they undergo. Under temperature stress conditions, the tertiary structure of molecules such as enzymes and other functional proteins can be damaged; they will not function properly and may be degraded. Fungi may produce heat or cold marker molecules such as heat shock proteins and chaperones to assist in the repair of functional structure (*Bai et al., 2015*; *Tiwari, Thakur & Shankar, 2015*). There will be substantial physiological changes that can be understood from their molecular-level responses. Such physiological changes can include the production of anti-freeze proteins coating the cell wall (*Bagwell & Ricker, 2019*), the production of extracellular hydrolases such as chitinolytic enzymes (*Fenice, 2016*), and increased production of heat shock proteins to minimize protein misfolding (*Kroll, Pahtz & Kniemeyer, 2014*; *Miteva-Staleva et al., 2017*). Such stress tolerance strategies improve cell and organism survival in stressed and sub-optimal environments (*Williams et al., 2011*).

Changes in protein contents generate a signature that can be identified and characterized. Potentially, these signatures can help understand organismal responses to stress, and be used as biomarkers for monitoring environmental changes and their effects on the ecosystem. Comparative analyses of fungal proteomic profiles can be used to understand changes in protein abundance that may serve as important biomarkers of temperature stress (*Kroll, Pahtz & Kniemeyer, 2014*). To date, most studies of fungal proteomic response towards temperature stress have been conducted on pathogenic species. Examples include the

black yeast, *Exophiala dermatitidis*, that can cause serious skin infections in humans (*Tesei et al., 2015*); *Aspergillus flavus*, a producer of lethal aflatoxins (*Bai et al., 2015*; *Delgado et al., 2015*); *Aspergillus fumigatus*, a thermo-tolerant human pathogen (*Albrecht et al., 2010*); *Ustilago maydis*, a basidiomycete that causes corn smut disease (*Salmeron-Santiago et al., 2011*); and *Metarhizium acridum*, an entomopathogenic species that can be used as an alternative to chemical insecticides (*Barros et al., 2010*). Several studies have also been carried out on extremophiles such as *Cryomyces antarcticus* in order to understand extreme stress tolerance from the proteomic perspective (*Isola et al., 2011*; *Tesei et al., 2012*; *Zakharova et al., 2014*).

In the last 10 years an increasing number of studies have analysed fungal temperature stress responses using proteomic approaches (*Albrecht et al., 2010*; *Bai et al., 2015*; *Salmeron-Santiago et al., 2011*; *Tesei et al., 2015*; *Zakharova et al., 2014*). Some have also used 'multi-omic' approaches that combine the simultaneous analysis of different molecules (proteins, mRNA, metabolites) to better understand complex biological responses (*Bai et al., 2015*; *Su et al., 2016*). Since the late 1990s, proteomic methodologies and technologies have evolved from an approach that relied on two-dimensional gel electrophoresis to gel-free analyses. The protein extraction method is a fundamentally important stage in any proteomic analysis. Optimisation of protein extraction is key in maximising yield and resolution, followed by the selection of proteomic profiling methods (*Bianco & Perrotta, 2015*; *Daim et al., 2015*; *Isola et al., 2011*). Label-free or labelled proteomic approaches, such as isotope and fluorescent labelling, are now used extensively because of their many advantages and provide useful information on differential expression of proteins in complex biological samples (*Fricker, 2018*). However, proteomic remains a very costly field, and interpreting the data obtained requires advanced knowledge of the entire technical process involved, as well as understanding of appropriate statistical principles for estimation and inference (*Karpievitch et al., 2010*).

## SURVEY METHODOLOGY AND OBJECTIVE

This review was developed based on research article journals published between 2008 and 2020, indexed in the Web of Science Core Collection and Scopus, Elsevier databases. We focus on studies of fungal response towards temperature stress elucidated using proteomic approaches. The effects of temperature stress (high and low) on metabolic pathways and HSPs specifically are a primary focus because, these two components are crucial in energy regulation and protein turnover, and hence for cell survival. This review provides an overview on the effects of temperature stress, on fungal proteomes and the implications this has for fungal responses within the ecosystem to current climate change challenges. The majority of the identified literature documents studies of filamentous fungi, although there are exceptions studying other groups, in particular the microcolonial rock-inhabiting fungus *C. antarcticus* and the black yeast *E. dermatitidis*.

### High temperature stress

In general, definitions of high temperature stress have primarily been made based on the optimum growth temperature ($T_{opt}$) and maximum growth temperature ($T_{max}$) (*Su et al.,*

*2016*; *Tesei et al., 2012*). However, this is not always the case. For example, in the study of pathogenic fungi, experimental temperature selection is often based on the host's body temperature (*Tesei et al., 2015*). Temperature can also be selected based on the induction or optimisation of mycotoxin production (*Bai et al., 2015*; *Salmeron-Santiago et al., 2011*). Deviations from the optimum temperature are expected to trigger a response at the molecular level, with the fungal proteome altering to prevent damage to the cell. Another important parameter in high temperature stress studies is the total duration or interval of heat exposure. Shorter exposures of seconds or minutes can lead to different proteomic expression compared to longer, or repeat, exposures (*Albrecht et al., 2010*; *Tesei et al., 2015*). This is because native proteins take time to undergo the processes of denaturing, unfolding or refolding, or eventually to be degraded under high temperature conditions (*Aragno, 1981*). There are two important stages in understanding the impact of high temperature stress on cells: the onset of any stress response, and its upper thermal limit. Together, these often give an indication of specific temperature stress response pathways. With an increase in environmental temperature, a higher metabolic rate is generally anticipated in cells. Thus, heat response pathways will be initiated to utilize the extra energy that is being produced from an increase in metabolic rate, and proteomic profiling can be used to understand the physiological changes taking place (*Richter, Haslbeck & Buchner, 2010*). However, on average, cells are only able to tolerate an increase in temperature of a few degrees above $T_{opt}$ (*Aragno, 1981*). Further increase in temperature beyond this limit causes irreversible damage and ultimately leads to cell death.

Table 1 summarises significant findings on fungal proteome profiling in response to high temperature stress, including the identification of proteins and suggestion of pathways that are potentially related to the identified proteins. A recent integrated -omics study of *Mrakia psychrophila* provided insight into the adaptation mechanisms of psychrophilic fungi (*Su et al., 2016*). At 4 °C and 20 °C, *M. psychrophila* produced an increased amount of major facilitator superfamily (MFS) proteins that are involved mainly in energy metabolism, compared to the level at 12 °C (optimal growth temperature range 12 °C−15 °C). However, heat shock proteins were upregulated only at 20 °C, which likely leads to the activation of the unfolded protein response (UPR). Consistent with this, at 20 °C, there was a negative correlation between the protein level change and transcript level change. The proteome abundances of another psychrophilic fungus, *Friedmanniomyces endolithicus* (optimal growth temperature range 10 °C−15 °C), assessed using a classical 2D gel electrophoresis (non-comparative) approach, showed a reduction from 284 (at 28 °C) to 224 (comparison between 15 °C and 28 °C) protein spots (*Tesei et al., 2012*). The authors hypothesized that, with exposure to high temperature, the basic set of proteins necessary to survive is relatively stable without the help of various chaperones. It is also likely that other non-protein protective metabolites and molecules are involved in the response pathways (*Keller, 2019*).

For human pathogenic fungi, their pathogenic characteristics become significant at body temperature. These have been explored in many species, such as *Penicillium marneffei* and *A. flavus*. *Gauthier (2017)* reviewed the molecular strategies used by thermally dimorphic fungi, focusing on their ability to adapt to core body temperature (37 °C) and transition

Abu Bakar et al. (2020), *PeerJ*, DOI 10.7717/peerj.10423

**Table 1** Summary of fungal proteomic profiling analysis when exposed to high temperature stress (work published from 2008–2020).

| $T_{exp}$ (°C) | $T_{opt}$ (°C) | Time | Species | # no. of proteins | | Proteins identified | Classification/ Pathways involved | Refs. |
|---|---|---|---|---|---|---|---|---|
| 20 °C | 12 °C | 1 month | *Mrakia psychrophila*[a] | 1673 | Down | Citrate synthase, DLST, LSC2 MTCP1 CLD1 | TCA cycle Oxidative phosphorylation Glycerophospholipid metabolism | *Su et al. (2016)* |
| | | | | | Up | DNAJ MPSY4181, MPSY2821 | Heat shock proteins MFS transporters | |
| 28 °C | 15 °C | 7 days | *Friedmanniomyces endolithicus*[b] | 284 | Down | Reduced no. of protein spots (141) | *Reduction in total number of protein spots, indicating a lack of a heat shock response | *Tesei et al. (2012)* |
| 37 °C | 25 °C | 1 day | *Penicillium marneffei*[b] | 270 | Up | Hsp 30, Hsp 70, antigenic mitochondrial protein HSP60 hypothetical protein (activator of HSP90 ATPase) succinyl-CoA synthetase alpha subunit, beta subunit of ATP synthase, NAD-dependent formate dehydrogenase adenylate kinase, phosphoglycerate kinase Ran GTPase spi1 Cdc48p UDP-N-acetylglucosamine pyrophosphorylase | Heat shock proteins Energy production and metabolism Regulate mitotic activities Cell cycle and division Synthesis of N-acetylglucosamine (monomeric component of cell-wall chitin) | *Chandler et al. (2008)* |

Peer J

**Table 1** (*continued*)

| T$_{exp}$ (°C) | T$_{opt}$ (°C) | Time | Species | # no. of proteins | | Proteins identified | Classification/ Pathways involved | Refs. |
|---|---|---|---|---|---|---|---|---|
| 37 °C | 28 °C | 1.5 days | *Aspergillus flavus*[a] | 3,886 | Down | 2-heptaprenyl-1,4-naphthoquinone methyltransferase, esterase/lipase, predicted O-methyltransferase, aminotransferase GliI-like, and aflE | Translation and biosynthetic pathways | *Bai et al. (2015)* |
| 37 °C | 28 °C | 2 h | *Ustilago maydis*[b] | – | Up | Enolase, Phosphoglycerate kinase Mitochondrial HSP 70, HSP 70, HSP 60 $\beta$-succinil CoA synthetase Glutathione S-transferase V/A- ATPase-A Oxidoreductase (AKR's) | Carbohydrate metabolism Protein folding Amino acid metabolism Redox regulation Ion homeostasis Other | *Salmeron-Santiago et al. (2011)* |
| 40 °C | 28 °C | 7 days | *Exophiala jeanselmei* *Coniosporium perforans*[b] | 174 255 | Down Down | Reduced no. of protein spots (208) Reduced no. of protein spots (70) | [*]Spots from the same pI (5–7) and molecular weight range (30-90 kDa) were extinct after heat stress, suggesting that both probably downregulated similar sets of proteins. | |
| | | | *Penicillium chrysogenum*[b] | 601 | Up | Increased no. of protein spots (220) | [*]Over-expression of proteins, interpreted as the synthesis of HSPs | *Tesei et al. (2012)* |

Abu Bakar et al. (2020), *PeerJ*, DOI 10.7717/peerj.10423

**Table 1** (*continued*)

| T_exp (°C) | T_opt (°C) | Time | Species | # no. of proteins | | Proteins identified | Classification/ Pathways involved | Refs. |
|---|---|---|---|---|---|---|---|---|
| | | | | 61 | Up | Calmodulin 2-Methylcitrate synthase mitochondrial, Type 1 phosphatases regulator YPI1 Putative aryl-alcohol dehydrogenase AAD6, Phenylalanine ammonia-lyase (fragment), Peptidyl-prolyl cis-trans isomerase ATP-dependent RNA helicase ded1, Elongation factor 1-alpha, Enolase 2, Fructose-2,6-bisphosphatase HSP 60 mitochondrial, HSP SSA2, HSP82, hsp98 | Developmental process Organelle part Catalytic activity Binding proteins Heat shock proteins | *Zou et al. (2018)* |
| 40 °C | 28 °C | 48 h | *Pleurotus ostreatus*[a] | 70 | Down | Subtilisin-like protease CPC735_066880 ATP-dependent RNA helicase DHH1, Myosin-1, Serine/threonine-protein kinase ppk8, Tubulin beta chain | Catalytic activity Binding proteins | |

**Table 1** (*continued*)

| T$_{exp}$ (°C) | T$_{opt}$ (°C) | Time | Species | # no. of proteins | | Proteins identified | Classification/ Pathways involved | Refs. |
|---|---|---|---|---|---|---|---|---|
| 48 °C | 30 °C | 0–4 min, Time interval (heat shock) | *Aspergillus fumigatus*[a] | 1886 | Up | HSP 70 chaperone HSP88 Tubulin alpha-1 subunit, actin-bundling protein Sac6 Nuclear movement protein NudC Allergen Asp F3 Transketolase TktA, hexokinase Kxk, Adenosylhomocysteinase, Nitrite reductase NiiA Cdc48 | Heat shock protein Cell wall and cytoskeleton Transport protein Defence against oxidative and nitrosative stress Carbohydrate and nitrogen metabolism Cell cycle and division | |
| | | | | | Down | Mitochondrial co-chaperone GrpE CRAL/TRIO domain protein Cytochrome c oxidase polypeptide vib | Chaperone Transport protein Energy generation | *Albrecht et al. (2010)*) |
| 50 °C | 30 °C | 24 h | *Aspergillus niger* 3.316[a] | 481 | Up | phosphatidylinositol 4-kinase type II subunit alpha, $\beta$-galactosidase, carboxypeptidase $\beta$-1,6-glucanase, extracellular $\alpha$-glucosidase rhamnogalacturonate lyase A | Cellular signalling Carbohydrate metabolism Cell wall organisation | *Deng et al. (2020)* |

**Notes.**

T$_{exp}$, Temperature of exposure; T$_{opt}$, Temperature of optimal growth.

Proteomic profiling approaches.

[a]Gel- free methods

[b]Gel-based method

*Non-conclusive remarks (no proteins identification involved).

to yeast morphology for virulence capabilities. These fungi used various strategies, such as upregulation of virulence factors, to promote cell adhesion to host, lysis of macrophages, avert cytokine responses, and impair host immunity. *Penicillium marneffei* is the only dimorphic species in its genus and forms its secondary cellular development, which is a uninucleate yeast, at 37 °C (*Chandler et al., 2008*). Increased levels of RanA expression, a GTP-binding nuclear protein that plays a role in nucleocytoplasmic transport, suggest that there is an additional signalling mechanism involved during phase transition in *P. marneffei*. Comparative work on extracellular proteomes of *P. marneffei* in yeast and mycelial phases showed upregulation of glyceraldehyde-3-phosphate dehydrogenase (GAPDH) and heat shock protein 60 (HSP60), respectively, that may play an important role in cell-host adherence (*Lau et al., 2013*). *Bai et al. (2015)* used an integrated transcriptomic and proteomic approach to study the response towards temperature changes in *A. flavus*, a species whose growth optimum is 37 °C, but that produces highest levels of mycotoxin is at 28 °C. The data obtained showed that a subset of 664 proteins involved in translation-related pathways, metabolic pathways and biosynthesis of secondary metabolites were differentially expressed at the two temperatures. At 28 °C, the expression pattern of proteins and transcripts related to aflatoxin biosynthesis were upregulated, and the authors suggested that change in the *aflR* transcript level was a better marker for the activation of aflatoxin biosynthesis. However, they also noted that there was a low correlation between overall transcript level and protein concentration in *A. flavus*, suggesting that the post-transcription modification processes may play a critical role in the regulation of the final protein expression level.

The plant pathogen, *U. maydis* is a well-studied causative agent for corn smut disease, but there are also reports of *Ustilago* spp. infections in humans (*McNeil & Palazzi, 2012*; *Teo & Tay, 2006*). Under various stress conditions, fungi often accumulate trehalose and initiate trehalose biosynthesis pathways (*Al-Bader et al., 2010*; *Nwaka & Holzer, 1997*). In a study of the effect of temperature stress on *U. maydis*, increased production of 11 proteins, commonly up-regulated in response to osmotic and sorbitol stress, was observed (*Salmeron-Santiago et al., 2011*). However, there were no changes in the trehalose concentration, thus suggesting that the up-regulated proteins are common proteins in the general stress response of *U. maydis*, non-specific to trehalose metabolism pathways. A more recent review (*Perez-Nadales et al., 2014*) highlighted the overall mechanisms of pathogenesis and some unifying themes among various fungal model organisms, emphasising the importance of conserved signalling pathways such as the cyclic adenosine monophosphate (cAMP)-dependent protein kinase A and mitogen-activated protein (MAP) kinases, and the central roles of secondary metabolic pathways in a very wide range of pathogenic fungi. However, comparison of the effect of temperature on the pathogenicity of fungal model organisms was not a focus of that study.

Mesophilic fungi, which have $T_{opt}$ of 25—30 °C, are generally able to tolerate a wider range of temperatures and associated stress, having a relatively wider range of temperature-dependent growth curves (*Dix & Webster, 1995*). In a non-comparative experiment on three mesophilic fungi (*Exophiala jeanselmei, Coniosporium perforans* and *Penicillium chrysogenum*), high temperature exposure led to variation in their proteomic expression

patterns (*Tesei et al., 2012*). Exposure to 40 °C increased the number of expressed proteins detected in *P. chrysogenum* but decreased those of *E. jeanselmei* and *C. perforans*. - A comparison was also made between species at 40 °C by selecting *P. chrysogenum* as the reference strain for mesophilic black fungi, which revealed 50 and 62 common protein spots in *E. jeanselmei* and *C. perforans*, respectively. The study suggested that there was a lack of a heat-shock protein response in *E. jeanselmei* and *C. perforans*. Furthermore, the disappearance of spots from the same pI and molecular mass range (respectively 5–7 and 30–90 kDa) suggested that both strains probably downregulated similar sets of proteins. *Tesei et al. (2012)* hypothesized that either the basic set of proteins that is important in high temperature exposure is stable without the help of heat-shock proteins or that other non-protein protective metabolites are involved. It is important to understand the limitations of such non-comparative experiments conducted through gel-based methods, where technical variations between replicates are expected. Quantification and identification of protein spots can be a challenge if differences in staining occur between two gels with no normalization.

An integrated analysis of transcriptomic and proteomic responses of *A. fumigatus* exposed to high temperature stress improved understanding of the thermotolerant characteristics responsible for the pathogenicity of this fungus (*Albrecht et al., 2010*). The study identified 91 differentially regulated protein spots representing 64 different proteins using two-dimensional difference gel electrophoresis (2D-DIGE) and tandem mass spectrometry (MS/MS) identification methods. These included a number of previously undescribed putative targets for the heat shock regulator Hsf1, providing evidence for Hsf1-dependent regulation of mannitol biosynthesis, translation, cytoskeletal dynamics, and cell division in *A. fumigatus*. *Albrecht et al. (2010)* also demonstrated a negative correlation between protein expression and gene transcription levels in *A. fumigatus* when exposed to supra-optimal temperatures (from 30 °C to 48 °C). The synthesis of most proteins was delayed by 60 to 90 min (medium delay) and up to 105 min (strong delay), after gene transcription by exposure to supra-optimal temperatures.

High temperature exposure causes specific stress response mechanisms to be activated that can be further explored using fungal proteomic profiles. Most fungi respond to high temperature stress by increasing the production of heat-shock proteins and chaperones and initiating alternative metabolic pathways (*Albrecht et al., 2010*; *Bai et al., 2015*; *Su et al., 2016*). However, some species show no significant changes in their proteome profiles, suggesting that the basic set of proteins needed for survival are highly thermotolerant (*Tesei et al., 2015*; *Tesei et al., 2012*). It has also been hypothesized that this is explained by the production of secondary metabolites and other non-protein protective metabolites that cannot be determined through proteomic approaches (*Zhang et al., 2016*). In addition, high temperature stress may induce a negative correlation between gene transcription and protein production in fungal cells due to post-translational modification and the initiation of protein degradation pathways, such as the ubiquitin-mediated protein degradation pathway and spliceosome-mediated decay (SMD) pathway (*Albrecht et al., 2010*; *Bai et al., 2015*).

## Low temperature stress

The study of low temperature stress resistance in microorganisms and other organisms has led to many beneficial findings. Proteins, such as cold-adapted enzymes, have been applied in many biotechnology industries such as in the manufacture of food and beverages, detergents, textiles, and in industrial molecular biology (*Sarmiento, Peralta & Blamey, 2015*). Research on cold-adapted proteins often involves psychrophilic or psychrotolerant microorganisms from the polar regions or deep oceans (*Cavicchioli, Thomas & Curmi, 2000*). Psychrophiles produce cold-adapted proteins with unique characteristics that have weak protein interactions, low thermal stability and increased specific activity, in order to achieve higher protein activity and flexibility at low temperatures (*Reed et al., 2013*). Protein structures are modified with subtle changes in the amino acid composition, thus remaining functional under extremely cold conditions. Wang et al. (2017) reviewed fungal adaptation to cold stress, providing an overview of life history strategies and highlighting the importance of cold-adapted fungi in the discovery of novel secondary metabolites and enzymes. Studies on fungal proteome profiles in response to low temperature stress are summarised in Table 2.

In a study of black rock-inhabiting fungi exposed to low temperature stress, their proteomic profiles generally showed an increase in total number of spots, except for *P. chrysogenum* (*Tesei et al., 2012*). *Friedmanniomyces endolithicus* and *C. perforans* proteome profiles exhibited changes in high molecular mass protein spots (70–170 kDa), while that of *E. jeanselmei* showed pattern changes at slightly lower molecular mass (25–100 kDa). *Penicillium chrysogenum* exhibited a decrease in total protein spots, interpreted as being due to lower metabolic activity under cold stress. *Exophiala dermatitidis*, a mesophilic fungus that is pathogenic in humans, was experimentally exposed to low temperature in order to understand the relationship between its thermotolerance properties and pathogenicity (*Tesei et al., 2015*). The strain was exposed to 1 °C (low temperature stress) for one week and comparison was then made with proteome profiles obtained at 37 °C (optimum temperature for growth) and 45 °C (high temperature stress). Using 2D-DIGE and a nano-scale liquid-chromatography electrospray-ionisation, tandem mass spectrometry (nLC-ESI-MS/MS), the study showed an average of 1,700 protein spots detected in *E. dermatitidis*. Exposure to low temperature stress led to a reduction in proteins associated with metabolic activity, mostly relating to general carbon metabolism. A large set of proteins involved in energy metabolism pathways were down-regulated, such as malate synthase, malate dehydrogenase, acetyl-coenzyme A synthetase and glyceraldehyde-3-phosphate dehydrogenase. Decreased levels of proteins involved in the response to heat stress such as Hsp70s, elongation factor 1α and Hsp30 might also be related to the reduction in metabolic rate and associated reduced energy consumption. Despite utilising alternative metabolic pathways, *E. dermatitidis* undergoes downregulation of metabolic pathways under exposure to non-optimal temperatures, resulted in a much slower growth rate. Comparison of the proteome profiles of *E. dermatitidis* under high and low temperature stress demonstrated the contribution to its success as a pathogen made by regulating the expression of basic thermotolerance proteins and modulating the production of proteins in major metabolic pathways.

Abu Bakar et al. (2020), *PeerJ*, DOI 10.7717/peerj.10423

**Table 2  Studies on fungal proteomic profiling in response to low temperature stress (published from 2008–2020).**

| T$_{exp}$ (°C) | T$_{opt}$ (°C) | Time | Species | # no. of proteins | | Proteins identified | Pathways/functions involved | Refs. |
|---|---|---|---|---|---|---|---|---|
| 1 °C | **15 °C** | 1 week | *F. endolithicus* | 466 | – | Increased no. of protein spots (41) | [*]Increased in high molecular weight spots (range from 70 to 170 kDa), suggesting production of cold-acclimation proteins | *Tesei et al. (2012)* |
| 1 °C | **28 °C** | 1 week | *E. jeanselmei* *C. perforans* *P. chrysogenum* | 387 494 358 | – | Increased no. of protein spots (5) Increased no. of protein spots (169) Decreased no. of protein spots (23) | [*]Changes in expression patterns (spots with MW of 25–100 kDa) [*]exhibited high molecular weight spots (range from 70 to 170 kDa), suggesting [*]production of cold-acclimation proteins slight decreased indicates downregulation of metabolic activity | |

**Table 2** (*continued*)

| T$_{exp}$ (°C) | T$_{opt}$ (°C) | Time | Species | # no. of proteins | | Proteins identified | Pathways/functions involved | Refs. |
|---|---|---|---|---|---|---|---|---|
| 1 °C | **37 °C** | 1 week | *E. dermatitidis* | 1,700 | Up | 14-3-3 family proteins Minor allergen Alt a7 Nucleoside diphosphate kinase Glyceraldehyde-3-phosphate dehydrogenase | Signalling proteins Small allergen molecules ATP production pathway Glycolytic pathway | |
| | | | | | Down | Acetyl-coenzyme A synthetase Alcohol oxidase Aldehyde dehydrogenase Phosphoenol pyruvate carboxykinase [ATP] Malate synthase, glyoxysomal | Acetate metabolism Alcohol metabolism Aldehyde metabolism Gluconeogenesis Pyruvate metabolism | |
| 1 °C | **45 °C**[**] | 1 week | *E. dermatitidis* | 1,700 | Up | 14-3-3 family proteins Acetyl-coenzyme A synthetase Minor allergen Alt a7 Nucleoside diphosphate kinase Glyceraldehyde-3-phosphate dehydrogenase Hsp30 | Signalling proteins Acetate metabolism Small allergen molecules ATP production pathway Glycolytic pathway Heat shock proteins | *Tesei et al. (2015)* |
| | | | | | Down | Alcohol oxidase Aldehyde dehydrogenase Phosphoenol pyruvate carboxykinase [ATP] Beta-lactamase Hsp70-like protein | Alcohol metabolism Aldehyde metabolism Gluconeogenesis Antibiotic resistance proteins Heat shock proteins | |
**Table 2** (*continued*)

| T$_{exp}$ (°C) | T$_{opt}$ (°C) | Time | Species | # no. of proteins | | Proteins identified | Pathways/functions involved | Refs. |
|---|---|---|---|---|---|---|---|---|
| 4 °C | **12 °C** | NA | *M. psychrophila* | 1,673 | Up | GLNA MPSY protein | Amino acid metabolism MFS transporter | *Su et al. (2016)* |
| | | | | | Down | citrate synthase and LSC2 proteins PDC | TCA cycle Glycolysis | |
| 12–15 °C | **23–25 °C** | 2 weeks | *Flammulina velutipes* | 1198 | Up | Trehalase (Fragment) Proteasome subunit $\beta$ type | Trehalose metabolism Proteosome pathway | *Liu et al. (2017)* |
| | | | | | Down | 60s ribosomal protein Acetylornithine aminotransferase Glutathione-disulfide reductase Adenylosuccinate synthetase 1 | Heat shock proteins Urea cycle Oxidative stress pathway Purine biosynthesis | |
| 12–15 °C | **23–25 °C** | 3 days | *Flammulina velutipes* | 1198 | Up | Heat shock cognate 70 Catalase | Heat shock proteins Catalytic enzymes | |
| | | | | | Down | Mitochondrial cytochrome T-complex protein eta subunit (Tcp-1-eta) Serine hydroxymethyltransferase Transketolase Methylmalonate-semialdehyde dehydrogenase | Electron transport chain Chaperonins Serine metabolism Pentose phosphate pathway Amino acid metabolism | |

Abu Bakar et al. (2020), *PeerJ*, DOI 10.7717/peerj.10423

Peer J

**Table 2** (*continued*)

| T$_{exp}$ (°C) | T$_{opt}$ (°C) | Time | Species | # no. of proteins | | Proteins identified | Pathways/functions involved | Refs. |
|---|---|---|---|---|---|---|---|---|
| 15 °C | **30 °C** | 24 h | *Mortierella isabellina M6-22* | 1,800 | Up | fructose-bisphosphate aldolase cytochrome c oxidase polypeptide 5B ATP synthase subunit $\beta$, ATP synthase $\delta$ subunit E3 ubiquitinprotein ligase BRE1, and histone acetyltransferase GCN5 | Glycolytic pathway Electron transport ATP production Ubiquitin–proteasome pathway | *Hu et al. (2016)* |

**Notes.**

  *Non-conclusive remarks (no proteins identification involved)

  **45 °C is not the optimal growth temperature for *E. dermatitidis*. However, it is one of the experimental temperature stress (high temperature stress)
*Mrakia psychrophila*, another psychrophilic yeast, showed a slightly different response to low temperature stress compared to *E. dermatitidis*, but more similar to other cold-adapted fungi such as *Flammulina velutipes* (*Liu et al., 2017*) and *Mortierella isabellina* (*Hu et al., 2016*). The responses included desaturation of fatty acids and accumulation of glycerol (*Su et al., 2016*). At 4 °C, two proteins were upregulated (glutamine synthetase, GLNA and MFS transporter protein, MPSY), and four proteins were downregulated (citrate synthase, succinyl co-A ligase, pyruvate decarboxylase and ribosomal protein). Comparative transcriptomic analysis showed that genes involved in ribosome production and energy metabolism were also upregulated at 4 °C. Proteomic analysis indicated that protein levels were positively correlated with transcription levels in some pathways under low temperature stress, suggesting the upregulation of chaperones and energy metabolism pathways. A study of the white rot fungus, *F. velutipes*, showed a similar change in protein profile in response to short-term and long-term exposure to low temperature (*Liu et al., 2017*). Of 63 differentially expressed proteins, 31 were upregulated, 24 of which were involved in energy metabolic pathways, amino acid biosynthesis and metabolism, signalling pathways, transport and translation. Four upregulated proteins were involved in energy metabolic pathways such as starch and sucrose metabolism (catalase, glucose-6-phosphate isomerase, trehalase and beta-glucosidase). However, catalase was upregulated only in the short term after exposure to low temperature, and its levels returned to normal after longer exposure to low temperature. Eleven differentially expressed proteins were involved in the biosynthesis of nine amino acids, indicating a role played in cold stress response by modifying nitrogen-containing molecule storage. Signalling molecules and processes involved in protein degradation were also upregulated, thus suggesting the importance of these molecules in controlling *F. velutipes* mycelium growth and fruiting body formation under cold stress. Proteomic profiling of *M. isabellina* (a soil fungus involved in many biotechnological applications) under low temperature stress identified 44 differentially expressed proteins under cold stress (*Hu et al., 2016*). These proteins were mainly involved in the regulation of ATP synthesis (ATP synthase subunit beta, ATP synthase d subunit), glycolytic pathways (fructose-bisphosphate aldolase), protein modification and electron transport (cytochrome c oxidase polypeptide 5B). The responses identified in *M. isabellina* also supported protein degradation pathways playing an important role in cold stress responses in fungi. Proteins such as HSPs and transitional endoplasmic reticulum ATPase (TER ATPase), that are essential in proteasome pathways, were also upregulated under cold stress.

Integrating the findings of these different studies has led to deeper understanding of cold-adaptive mechanisms in fungi. They respond to low temperature stress by modifying their energy metabolic pathways, increasing the production of chaperones to minimise misfolded protein production, and regulating the synthesis of amino acids that are important in nitrogen storage. In contrast with studies of high temperature stress, the overall response of many fungal species indicates that low temperature does not cause irreversible damage to cells, with responses acting by modifying molecular content rather than wholesale transformation of complex protein networks. The huge amount of information that can be generated in fungal proteomic studies examining low temperature stress can be utilised

for more specific approaches, for example in process optimization and purification of cold-adapted biomolecules. Future studies on cold adaptation mechanisms in fungi will benefit from a focus on identifying common proteins that act as cold stress biomarkers and exploiting the advantages of cold-adapted proteins in biotechnological applications.

## Effects of temperature stress in fungal proteomes

Fungi respond to temperature stress through regulation of various proteins, that can be visualised using proteomic profiling techniques within the limitations noted above. Changes in fungal proteomes exposed to various temperature stresses (Tables 1 and 2) demonstrate that many cellular functions are affected, such as the TCA cycle and energy production and metabolism, oxidative phosphorylation, regulation of mitotic activities, cell cycle and division, transcriptional and translational stages of biosynthetic pathways, cell wall and cytoskeleton reorganisation, transport protein remodelling, defence against oxidative and other stresses, and cellular signalling mechanisms. Undeniably, these proteins and their roles in cellular functions are highly significant, with the complex cellular protein network maintained through protein homeostasis (*Mühlhofer et al., 2019*; *O'Neill et al., 2020*). It is extremely challenging to identify any specific set of proteins as being the most affected by or vulnerable to temperature stress, or being used as potential temperature stress biomarkers. Metabolic pathways and HSPs are considered a primary focus and elaborated below. These two components are crucial in energy regulation and protein turnover, and hence for cell survival (*Yan et al., 2020*). More widely, biosynthesis and regulation of metabolic pathways and HSPs are of importance in many applied fields such as agriculture, food biotechnology and medicine (*Ene et al., 2014*; *Lamoth, Juvvadi & Steinbach, 2016*; *Sarmiento, Peralta & Blamey, 2015*).

## Effect of temperature stress on metabolic pathways

Fungal adaptation towards temperature stress involves alterations in the utilisation of many metabolic pathways in trade-offs to compensate for the amount of energy needed or the extra energy to be utilised in delivering the stress response. As temperature increases, the rate of metabolism (of a non-endothermic organism) increases as a purely physical consequence, and then rapidly declines at higher temperatures as the metabolic systems start to fail—eventually leading to cell death. Metabolic regulation is one strategy employed by organisms to adapt to temperature stress, which is controlled by factors such as concentration of substrates, products or allosteric effectors (*Clarke, 2018*; *Suarez & Moyes, 2012*). Understanding thermal adaptation in fungi requires an overall understanding on the effects of temperature on pathway flux, including such things as the roles and limitations of enzymes involved, concentration of substrates and products, and other protein and non-protein components (*Schulte, 2015*). Advances in proteomic technologies have made it possible to determine many of the proteins involved in the complex network of metabolic pathways, as low as in zeptomole-scale mixtures (*Swaminathan et al., 2018*), but it remains challenging to accurately identify and allocate all protein molecules involved into their various pathways (*Timp & Timp, 2020*). However, many studies have developed visual diagrams of complex protein-protein interaction networks, and it is accepted that

temperature stress causes quantifiable changes in these metabolic pathways compared to non-stress conditions (*Bai et al., 2015*; *Kostadinova et al., 2011*).

Glycolysis is an important metabolic pathway in most organisms, including fungi. Many fungi produce enzymes to break down complex polysaccharides such as cellulose, hemicellulose, pectin and starch to produce glucose (*Krishnan et al., 2016*; *Xiong et al., 2017*). Many studies have confirmed that fungi produced a range of extracellular hydrolases to help them utilise different carbon and nitrogen sources under exposure to temperature stress (*Krishnan et al., 2017*; *Tajuddin et al., 2018*). The production of these enzymes responds to changing environmental temperature, in order to maintain metabolic requirements and protein turnover. As noted earlier, trehalose production is also important for fungi under temperature stress. Trehalose is known to be important in the acquisition of thermotolerance and desiccation tolerance in many fungal species (and much more widely across multiple groups of biota) (*Everatt et al., 2015*; *Gancedo & Flores, 2004*; *Liu et al., 2019*; *Tereshina, 2005*). In *Saccharomyces cerevisiae*, trehalose and intracellular water stabilise the membrane structure and other intracellular networks under temperature stress conditions (*Piper, 1993*). Various studies document increased activity of trehalases, such as the trehalose synthase and fructose-1,6-biphosphatase, in response to heat stress (*Bonini et al., 1995*; *Cai et al., 2009*). In addition, enzymes such as cAMP-dependent protein kinase and plasma membrane ATPase also play significant roles in fungal thermotolerance determination (*Jurick Ii, Dickman & Rollins, 2004*; *Piper, 1993*). Enolase, an enzyme that converts 2-phosphoglyceric acid to phosphoenolpyruvic acid in glycolysis, is also crucial in the heat stress response of many fungal species and yeasts (*Ji et al., 2016*). It has been suggested that enolase (ENO1) is closely involved in heat stress responses, and ENO1 of yeasts and streptococcal strains isolated from rats has high thermal stability (*Cuéllar-Cruz et al., 2013*; *Iida & Yahara, 1985*; *Kustrzeba-Wójcicka & Golczak, 2000*).

## Effect of temperature stress on heat shock proteins (HSPs)

In the event of heat stress, HSPs are produced in cells to protect proteins from aggregation, unfold or refold aggregated proteins, or target them for the degradation pathway. Although initially named following discovery in heat stress experiments, HSPs are a part of general stress responses, produced when cells and organisms are exposed to multiple types of stressors. Most HSPs are molecular chaperones produced constitutively or induced upon cell stress, which can be triggered by a temperature change of just a few degrees (*Richter, Haslbeck & Buchner, 2010*). Heat shock proteins are primarily categorised and named according to their molecular mass, which varies from 10 to 100 kDa. Their multifunctional properties have made them an important and reliable target biomarker in various fields such as crop management, plant and microbe adaptation towards environmental stress, and cancer related studies (*Jee, 2016*). HSPs are known to play a role in temperature-stressed environments (*Tiwari, Thakur & Shankar, 2015*), but understanding the complexity of the HSP network as a defence mechanism in the cellular environment requires much further research. The cellular functions of the various HSPs that are involved as part of the heat shock response network in fungi, using *S. cerevisiae* as a model, has been discussed in depth (*Verghese et al., 2012*). Evidence suggests that HSPs are involved in cell cycle arrest (*Vergés*

*et al., 2007*), metabolic reprogramming (*Elliott, Haltiwanger & Futcher, 1996*), modulating cell wall and membrane dynamics (*Shaner, Gibney & Morano, 2008*; *Truman et al., 2007*; *Winkler et al., 2002*) and protein aggregation (*Nathan, Vos & Lindquist, 1997*).

Heat shock protein 60 (HSP60) is a highly conserved molecule in many organisms and has been found to respond to various stress conditions in fungi. This HSP is upregulated in response to increased temperature in species such as *A. fumigatus*, *Aspergillus terreus*, *P. chrysogenum*, *Scedosporium apiospermum*, *Trichophyton mentagrophytes*, *Candida albicans* and *S. cerevisiae* (*Raggam et al., 2011*). HSP60 has also been proposed to play a role in the assembly of precursor polypeptides into oligomeric complexes following incorporation into the mitochondrial matrix (*Patriarca & Maresca, 1990*). *Galello, Moreno & Rossi (2014)* recently described the interaction of HSP60 and Ira2 with Bcy1, a regulatory subunit of protein kinase A (PKA) in *S. cerevisiae*. Using an MS-based proteomic approach, this study demonstrated that HSP60 localized the entire PKA-Ras complex to mitochondria under the regulation of the cAMP-PKA-signalling pathway. There are also reports of HSP70 having a role in enhancing fungal resistance to heat stress and other abiotic stresses (2010; *Montero-Barrientos et al., 2008*). The overexpression of HSP0 genes in *Trichoderma harzianum* T34 resulted in an increase in biomass and enhanced tolerance to other abiotic stresses after heat shock treatment (*Montero-Barrientos et al., 2008*). Subsequently, the function of the HSP70 gene from *T. harzianum* T34 was further studied in the transgenic plant, *Arabidopsis thaliana*, showing that the transgenic plants exhibited higher tolerance towards heat stress (*Montero-Barrientos et al., 2010*). HSP90, one of the most ubiquitous chaperones in yeasts (*Nathan, Vos & Lindquist, 1997*), was found to interact with calcineurin (*Imai & Yahara, 2000*) and respond to heat stress indirectly through facilitating the activation of the MAPK complex (*Truman et al., 2007*). More recent work has shown that HSP90 plays a central role in heat stress responses in *Fusarium graminearum*, in addition to its crucial roles in fungal vegetative growth, reproduction and virulence (*Bui et al., 2016*). In *A. fumigatus*, HSP90 has been implicated in drug resistance, as disruption of the HSP90 circuitry leads to activation of antifungal activity in caspofungin (*Lamoth, Juvvadi & Steinbach, 2016*).

## CONCLUDING REMARKS

The importance of fungi and of mycological studies in the development of the bioeconomy and in environmental monitoring has been accepted and highlighted by researchers worldwide. Excellent reviews are available documenting fungal response towards abiotic stress and how fungi can affect the tolerance of other organisms, especially plants, towards multiple stressors (*A'Bear, Jones & Boddy, 2014a*; *Coleman-Derr & Tringe, 2014*; *East, 2013*; *McCotter, Horianopoulos & Kronstad, 2016*; *Rangel et al., 2018*). Many studies have set out to understand fungal heat stress response mechanisms in order to help develop tools for crop monitoring and production as well as producing heat-tolerance species that are better able to response to climate change challenges (*Jagadish et al., 2010*; *Simova-Stoilova, Vassileva & Feller, 2016*). Research on fungal molecular responses towards temperature stress has been widely explored using different omics approaches. Studies on fungal genomes, transcriptomes, secretomes and proteomes have provided new knowledge and

understanding of these responses (*Albrecht et al., 2010*; *Bai et al., 2015*; *Cologna et al., 2018*; *Kroll, Pahtz & Kniemeyer, 2014*). The outcomes of these studies will need to be fully exploited to maximise the potential of fungi both in biotechnological industries and in natural environment under current climate change scenarios (*Classen et al., 2015*; *Lange et al., 2012*; *Rangel et al., 2018*).

As is clear from this review, there is currently limited available literature documenting the use of proteomics in studying the effects of temperature stress on fungal proteins, and most reports to date have focused on pathogenic fungi. Despite the tremendous promise of proteomics, technical limitations that affect sensitivity and resolution continue to limit its practical utility. In almost all studies considered here, detection was limited to medium and high abundance proteins, the majority of which were related to metabolism and chaperones. With currently available data from proteomics, it remains difficult to provide comprehensive descriptions of the biological changes that can be observed. At present this means our ability to identify specific biomarkers for any given change is very limited, as many confounding factors can influence the production of these more abundant proteins.

These issues can be overcome in future. More thorough interrogation of the proteome, including integration within multi-omics platforms, will allow for deeper insights into the mechanisms underlying changing cell physiology. If potential biomarkers are to be identified, proteomic analyses must go beyond medium and high abundance proteins. Several strategies can be employed to achieve this. Specific sub-proteomes of proteins and their associated pathways already identified as changing in initial analyses can be further explored. A targeted proteomics approach, for example looking at specific sub-fractions of the cell (cell wall, specific organelles, etc.) may be beneficial. This will allow for the identification of low abundance proteins that may include, for instance, signaling proteins with specific roles in heat stress response. Further information can also be obtained by conducting investigations using multi-omics platforms applied to the same samples simultaneously, to provide both complementary and confirmatory information.

## ACKNOWLEDGEMENTS

We also thank Prof. Peter Convey from the British Antarctic Survey (BAS), United Kingdom, for commenting on the manuscript.

### Funding

The study was supported by the Malaysian Ministry of Higher Education (MOHE) through their funding programme Higher Centre of Excellence (HiCoE) (grant number IOES-2014G), and the Universiti Malaya Research Programme (UMRP) (grant number RP026A-18SUS), the University of Malaya (UM) (OCAR TNC(P&I) 2011 Account No. (A—55001—DA000—B21520), and a postgraduate sponsorship from the Majlis Amanah Rakyat Malaysia (MARA Scholarship Programme). The funders had no role in study design, data collection and analysis, decision to publish, or preparation of the manuscript.

## Grant Disclosures

The following grant information was disclosed by the authors:

Higher Centre of Excellence (HiCoE): IOES-2014G.

Universiti Malaya Research Programme: RP026A-18SUS.

University of Malaya (UM) (OCAR TNC(P&I) 2011: A−55001−DA000−B21520.

Majlis Amanah Rakyat Malaysia (MARA Scholarship Programme).

## Competing Interests

The authors declare there are no competing interests.

## Author Contributions

- Nurlizah Abu Bakar conceived and designed the experiments, performed the experiments, analyzed the data, prepared figures and/or tables, authored or reviewed drafts of the paper, and approved the final draft.
- Saiful Anuar Karsani and Siti Aisyah Alias conceived and designed the experiments, authored or reviewed drafts of the paper, and approved the final draft.

## Data Availability

A traditional rather than a systematic review approach was taken to focus on fungal responses to temperature stress elucidated using proteomic approaches. Information gathered in this article is from journal search engines such as Scopus and Web of Science Core Collection.

There is no raw data because this is a literature review.

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
