# Peer review of "Fungal survival under temperature stress: a proteomic perspective"

_PeerJ, doi:10.7717/peerj.10423_

## Round 0.1 · original submission · Major Revisions

The manuscript has been reviewed by three experts in the field, and even though they find merits in the manuscript, some concerns have to be addressed before considering for publication in PeerJ.

In particular, the expansion to 2020 papers is highly recommended, along with the dimorphism point raised by one of the reviewers.

Moreover, this Editor thinks that the biological aspects addressed in the manuscript are extremely limited. The authors should expand the section metabolic pathways, to provide a thorough revision, to include changes in the secretory pathway, the cell wall remodeling, transcriptional and translation aspects, for example. Most importantly, a Review manuscript should gather the most relevant information on the specific subject and from this, to generate new hypotheses, points of view or ground-shaking comments to fuel up further discussion in the specific field. This is not included in the manuscript and the authors are encouraged to attend this point too.

·

Basic reporting

The review collects fungal proteomics work produced during the decade 2008-2018 with a special focus on proteome rearrangements as triggered by temperature stress. The manuscript is written in a clear and unambiguous English and is of interest, however there are some recommendations below to strengthen and improve its quality.

A number of reviews on fungal proteomics are available documenting stress response, host-pathogenic interactions, biotechnological applications or describing the advancement of proteomics techniques applied to the investigation of the fungal proteome. As compared to those, this manuscript offers an insight into the very specific topic of temperature stress, which had not been the main subject of a review on fungal proteomics yet. The introduction adequately explains the subject of the review, describing the crucial role of fungi in soil microbial communities as well as the importance of protein biomarkers of fungal stress for the monitoring of environmental changes.

Experimental design

The article content is within the aims and scope of the journal. The review collects the results of not more than 10 studies produced between 2008 and 2018, which reflects the limited number of proteomics studies specifically including temperature stress hitherto available. Two considerations on this regard: 1_ I understand the aim to focus on a specific time span, however I fear that excluding the last two years of research will have an impact on the completeness of the review, along with making it not thoroughly up-to-date. Two works, among others, that focus on heat stress, would be worth a mention i.e. Zou et al., 2018 DOI=10.3389/fmicb.2018.02368 and of Deng et al., 2020 https://doi.org/10.1002/mbo3.1012. 2_One of the aims of the manuscript is to provide an overview of the implications of temperature stress for fungal responses within the ecosystem to current climate change challenges. However, the authors do not expand on this aspect other than in the introduction, whose content sets expectations about a manuscript primarily dealing with soil fungi and their potential as biomarkers of global warming.

Validity of the findings

No comment.

Additional comments

Line 41: “global ice loss will”. Here the verb is missing.
Line 61: Please consider changing “play” with “plays”
Line 94: “have analysed on fungal temperature”. Please consider removing “on”.
Line 115-118: the authors state that the objective of the review is “to provide an overview on the effects of temperature stress on filamentous fungal proteomes”. Nonetheless, some of the species listed in the review, e.g. Cryomyces antarcticus (microcolonial RIF) and Exophiala dermatitidis (black yeast), are not exactly filamentous fungi. I would suggest clarifying this in the text.

Lines 145-159: This paragraph is part of a subchapter entitled “High temperature stress” and references to psychrophiles are made using M. psychrophile and F. endolithicus as examples. To be thorough and give the reader a better idea of the stress, I’d suggest adding the optimal temperatures of growth of both the yeast and the fungus to the text.

In the case of F. endolithicus the authors should provide more details. “The proteome abundances of another psychrophilic fungus, F. endolithicus, assessed using a classical 2D gel electrophoresis (non-comparative) approach, showed a reduction from 284 to 224 protein spots”. In relation to the exposure to what temperature did this occur?

Line 165: Please clarify in the text what RanA is, in order to make the statement “Increased levels of
RanA expression suggest that there was an additional signalling mechanism involved during phase transition of P. marneffei”, understandable.

Line 166-167: “Gauthier (2017) reviewed the molecular strategies used by thermally dimorphic fungi (Gauthier 2017).” The authors do not provide any details about the findings of the cited research. Please consider expanding on it.

Line 169: Please consider changing “a species whose growth is optimum at 37°C but maximum mycotoxin is produced at 28°C.” with: A species whose growth optimum is.., but highest levels of mycotoxin are produced at

Line 201-203: I’d suggest rephrasing this sentence and improving the wording
Lines 226-231: I suggest splitting this very long sentence in two parts.
Lines 247-248: “Protein structures are modified with subtle changes in the amino acid composition, thus remaining functional extremely cold conditions.” Here, the preposition (e.g. “under” or “upon”) is missing.
Line 248: Please use the past here.

Line 254-260: Please pay attention to when to use the extended or shorten species name. Generally, the full name should be used the first time that a species is mentioned in a text. In the second and subsequent uses of the name, the name can be abbreviated. Please check this throughout the text.

Line 271: Please consider changing “Decreased levels of heat shock proteins” with Decreased levels of proteins involved in the response to heat stress”.

Lines 273-275: The authors cite the work of Tesei et al (2015) stating the following: “Exophiala dermatitidis lacks a typical stress response under exposure to non-optimal temperatures, possibly because the cost of utilising alternative metabolic pathways may outweigh the benefits of much slower growth rates”. This sentence is not entirely clear. According to Tesei et al. (2015) a much slower growth rate is a consequence of the down-regulation of the metabolism that black fungi undergo upon low temperature stress.

Lines 280-281: The authors state that “Mrakia psychrophila has been shown to respond to low temperature stress in a very similar fashion to other cold-adapted fungi.” However, chaperones and energy metabolism pathways are found to be upregulated in the yeast upon exposure to low temperature, which is in contrast with what observed in E. dermatitidis, the study cited just before in the text. What do the authors mean by cold-adapted fungi? Psychrophilic fungi? Please consider specifying this in the text.

Line 289: Please consider writing the extended name of F. velutipes, since it is the first time that it is mentioned in the text.

Line 389: typo. “HSP60 has alos been”

Table 2: In this table, for each of the cited studies, both the temperature of exposure and the temperature of optimal growth of the fungus are listed. In the case of E. dermatitidis, both 37 and 45°C are indicated as T optimum, however the correct temperature optimum is solely 37 °C. In Tesei et al. (2015), following growth at 37°C, E. dermatitidis was exposed to 1 or 45°C for 1 hour (short-term exposure) or 1 week (long-term exposure) in order to monitor the yeast response to both low- and high-temperature stress. Protein profiles from the different temperatures of exposure were compared in order to assess protein modulation.
Please consider editing the table as well as the text, accordingly. The portion of Table 2 describing proteins and pathways affected by the exposure of E. dermatitidis (with optimum at 45°C) to 1°C shall be edited to point out that 45°C is not the fungus temperature optimum, but one of the experimental conditions.
Please also cross-check the part of Table 2 displaying up and down regulated proteins at 1°C as compared with the optimal temperature of 37°C, as e.g. acetate metabolism is listed among up-regulated pathways but instead it exhibits lower levels at 1°C.

·

Basic reporting

Overall well written and informative. Some minor grammatical errors (e.g., fungi proteomes in abstract should be fungal proteomes, various should be various). Good overview of relevance and importance of mycological studies and fungal organisms within communities. Language is passive at times (e.g., can be due to, been made based on the), such statements should be minimized to strengthen the impact of the manuscript. Some proof-reading of acronyms and shortening of scientific names before proper names are introduced need to be addressed.

Experimental design

The authors state previous work and reviews available and highlight the uniqueness of the current manuscript. The survey methodology is unbiased in coverage of the subject and is mostly comprehensive and refers to other reviews for research not covered in this manuscript. Sources are adequately cited.
Heat shock proteins are discussed in the introduction and their relevance and importance is presented but then they are not highlighted again until the last section. Perhaps move the section on the impact of temperature stress on HSP to after the introduction if they are important to the studies presented in the Review.

Validity of the findings

The studies are well reported and results are interpreted well with some suggestions for future work provided. Tables 1 and 2 are well-organized and helpful for the reader. Notes in the General Comments section address concern over arguments and focus established in the abstract and introduction that are not clear throughout the manuscript.

The concluding remarks section would be strengthened by highlighting specific information gained by conducting the current Review

Additional comments

The title refers to temperature stress in general, but the abstract emphasizes agricultural applications and the main body of the text focuses on agricultural and medical fungal species. Providing some clarity on the emphasis in the abstract would help the reader follow the direction of the manuscript. Why emphasizing agricultural relevance? It’s a good topic and important but a little more justification would help the reader.
Several of the highlighted studies seem to use qualitative analyses to determine in proteins are increased or decreased in abundance by reporting presence/absence. It would be valuable for the authors to discuss limitations of such studies where absence may be technical variation/error if quantification values are not available, or increased abundance could be attributed to differences in staining between two gels if no normalization is provided. Such statements could be provided in the last paragraph of the ‘high temperature stress’ section as this is paragraph reads as a discussion of the studies reported to date.

Specific comments:
Abstract:
- What is the difference between a traditional vs. systematic review process?
- How can PTM in stress pathways identify potential biomarkers? The connection is not evident, although later in the introduction the importance of profiling proteomes for biomarkers is clear but the connection to PTMs remain elusive.
Line 41: ice loss will ? (missing word)
Line 71: is there a reference for anti-freeze proteins (other points in this sentence had references)
Line 92: metabolic differences relevant to heat stress?
Line 94: ‘In the last 20 years…’ proteomics has been used to study fungal temperature stress and all references are within the past 10 years. Seems more accurate to state within the past 10 years. As, the methodology focuses on a 10 year period of study.
Line 97: In general, the field of proteomics was established in the late 90s, but the sentence states ‘over the past 4 decades’, I would argue that over the past two decades is more accurate.
Line 102: why highlight the use of isotopic tags? Many other approaches exist. Are these approaches generally used for heat stress proteomics? Why is pooling samples an advantage? This paragraph is informative but seems to jump around in focus.
Line 111: suggest defining what the ‘traditional approach’ is compared to other approaches. Why was a traditional approach selected?
Line 258: check usage of fungal names (e.g., Penicillium chrysogenum is spelled out by abbreviated in a couple lines above) same as in line 273, 354
Line 266: DIGE and nLC-ESI-MS/MS need to be spelled out
Line 337: What advances in proteomic technologies have been made? Reference to specific aspects would be helpful.
Line 369 HSP used previously, not need to be redefined here
Line 394: cAMP-PKA mentioned in above sections but only shortened in this paragraph. Same with MAPK.
Line 424: PTM discussed but not mentioned in the Review. Why is this relevant now? References to support?

Reviewer 3 ·

Basic reporting

Review article entitled ‘Fungal survival under temperature stress: A proteomic perspective’ uses a limited range of studies sketching the effect of temperature range on different fungi. The followings are the comments to improve the article.

Authors have considered temperature as a stress factor to compile the review; A temperature range is favorable for growth, and below/above of that rage is /are considered as stress. When the optimum/temperature range for growth has been shifted to a 'temperature' referred to as 'temperature stress', the author should discuss the criteria and to strengthen the manuscript discuss it with specific examples?

Experimental design

Table-1 indicated that 7 studies have been included for the basis of this review? A column needs to be put in for “favorable temperature range of growth”

Line 401-407 Hsp90 has also been implicated in drug resistance? Add this point?
Lamoth F, Juvvadi PR, Steinbach WJ. Heat shock protein 90 (Hsp90): A novel antifungal target against Aspergillus fumigatus. Crit Rev Microbiol. 2016;42(2):310-321. doi:10.3109/1040841X.2014.947239
Line 69-75 A recent review on temperature/cold stress/heat shock proteins has summarized well on fungal biology is missing in the introduction section or elsewhere?
Tiwari S, Thakur R, Shankar J. Role of Heat-Shock Proteins in Cellular Function and in the Biology of Fungi. Biotechnol Res Int. 2015;2015:132635. doi:10.1155/2015/132635

In dimorphic fungi ‘temperature’ is the key factor for transition/dimorphic switch of fungi from one to form to another form. Why the author did not consider those fungi e.g., Paracoccidioides, etc, where temperature stress/thermoregulated fungi will be addition/ relevant to the reader.

Concluding remarks; It misses the punch line or the outcome of the review should be brought in here to strengthen the review article.

Validity of the findings

No comments

Additional comments

As above

---

## Round 0.2 · Major Revisions

I have been asked to take over from the original editor, who is no longer available.

The authors have addressed most of the comments shared in the previous message but no effort was done to deal with the comments raised by this Editor, which are again given lines below.

This Editor thinks that the biological aspects addressed in the manuscript are extremely limited. The authors should expand the metabolic pathways section, to provide a thorough revision, to include changes in the secretory pathway, the cell wall remodeling, transcriptional and translation aspects, for example. Most importantly, a Review manuscript should gather the most relevant information on the specific subject and from this, to generate new hypotheses, points of view, or ground-shaking comments to fuel up further discussion in the specific field. This is not included in the manuscript and the authors are encouraged to attend this point too.

---

## Round 0.3 · accepted · Accept

The authors have significantly improved the manuscript content and it is now suitable for publication in Peer J.